

# The clinical association of programmed death-1/PD-L1 axis, myeloid derived suppressor cells subsets and regulatory T cells in peripheral blood of stable COPD patients

Mingqiang Zhang[1], Yinghua Wan[1], Jie Han[1], Jun Li[1], Haihong Gong[2] and Xiangdong Mu[1]

[1] Department of Respiratory and Critical Care Medicine, Tsinghua Changgung Hospital, School of Clinical Medicine, Tsinghua University, Beijing, China
[2] Affiliated Hospital of Qingdao University Medical College, Department of Respiratory and Critical Care Medicine, Qingdao, China

Corresponding author
Haihong Gong,
haihonggong@qdu.edu.cn

## ABSTRACT

**Background:** Myeloid-derived suppressor cells (MDSCs) have crucial immunosuppressive role in T cell dysfunction in various disease processes. However, the role of MDSCs and their impact on Tregs in COPD have not been fully understood. The aim of the present study is to investigate the immunomodulatory role of MDSCs and their potential impact on the expansion and function of Tregs in COPD patients.

**Methods:** Peripheral blood samples were collected to analyze circulating MDSCs, Tregs, PD-1/PD-L1 expression to assess the immunomodulatory role of MDSC and their potential impact on the expansion and function of Treg in COPD. A total of 54 COPD patients and 24 healthy individuals were enrolled in our study. Flow cytometric analyses were performed to identify granulocytic MDSCs (G-MDSCs), monocytic MDSCs (M-MDSCs), Tregs, and the expression of PD-1/PD-L1(L2) on MDSCs and Tregs in peripheral blood.

**Results:** Our results revealed a significantly higher percentage of G-MDSCs and M-MDSCs ($p < 0.001$) in COPD patients compared to the healthy controls. Additionally, a significantly higher proportion of peripheral blood Tregs was observed in COPD patients. Furthermore, an increased expression of cytotoxic T-lymphocyte-associated protein 4 (CTLA-4) on Tregs ($p < 0.01$) was detected in COPD patients. The expression of PD-1 on CD4$^+$ Tcells and Tregs, but not CD8$^+$Tcells, was found to be increased in patients with COPD compared to controls. Furthermore, an elevated expression of PD-L1 on M-MDSCs ($p < 0.01$) was also observed in COPD patients. A positive correlation was observed between the accumulation of M-MDSCs and Tregs in COPD patients. Additionally, the percentage of circulating M-MDSCs is positively associated with the level of PD-1 ($r = 0.51$, $p < 0.0001$) and CTLA-4 ($r = 0.42$, $p = 0.0014$) on Tregs in COPD.

**Conclusion:** The recruitment of MDSCs, accumulation of Tregs, and up-regulation of CTLA-4 on Treg in COPD, accompanied by an increased level of PD-1/PD-L1, suggest PD-1/PD-L1 axis may be potentially involved in MDSCs-induced the expansion and activation of Treg at least partially in COPD.

# INTRODUCTION

Chronic obstructive pulmonary disease (COPD) is a chronic lung disease, characterized by persistent flow limitation and aberrant systemic inflammation (*Brightling & Greening, 2019*; *Adeloye et al., 2022*). Smoking is one of the most well-established risk factors for COPD (*Riley & Sciurba, 2019*; *Adeloye et al., 2022*). A growing body of evidence suggests that autoimmune response plays a fundamental role in COPD pathogenesis and has garnered significant attention (*Caramori et al., 2018*; *Hou & Sun, 2020*). The imbalance between proinflammatory responses and immunosuppressive cells, such as MDSCs and Tregs, or cytokines to self-antigens may contribute to immune dysfunction and disease progression of COPD.

T cells are considered critical in controlling airway inflammation induced by cigarette smoke (*Caramori et al., 2018*). Regulatory T cells (Tregs) are responsible for inhibiting T cell activation and are essential for maintaining peripheral immune homeostasis (*Sakaguchi et al., 2020*). However, the precise role of Treg cells in the disturbed immune homeostasis of COPD has not been thoroughly investigated and the results of previous research are not entirely concordant (*Hou & Sun, 2020*). The number of Treg cells fluctuates in both COPD patients and animal models (*Demoor et al., 2010*; *Gong et al., 2017*; *Sales et al., 2017*). The impaired suppression of CD4$^+$Tcell activation and reduced IL-10 secretion may suggest the impaired function of Treg cells in COPD (*Hou & Sun, 2020*).

PD-1 and its ligands (PD-L1/PD-L2), which play a crucial role in regulating T-cell activation and Treg cell development, have garnered considerable attention (*Gianchecchi & Fierabracci, 2018*; *Adamczyk & Krasowska, 2021*). PD-L1 has been found to be more effective than PD-L2 in inhibiting T-cell activation and is particularly important in maintaining tissue tolerance (*Mi et al., 2021*). PD-1/PD-L1 axis primarily maintains immunologic homeostasis under normal circumstances and mediates immune escape during the development of tumors (*Cha et al., 2019*). PD-1-targeted antibody drugs have shown significant anti-tumor efficacy in certain solid tumors, further confirming their key role in the immune response. In contrast, studies on PD-1/PD-L1 in non-neoplastic diseases, including pulmonary inflammatory diseases, are relatively insufficient. Current research has demonstrated that PD-1/PD-L1 can be induced by inflammatory factors in chronic infectious diseases (*Chinai et al., 2015*). Additionally, peripheral blood lymphocytes from COPD patients expressed significantly higher levels of PD-1 (*McKendry et al., 2016*; *Tan et al., 2018*). However, the conclusions were inconsistent. Some scholars detected that the low level of PD-L1 in DCs of COPD patients might be part of the mechanisms promoting disease progression (*Stoll, Virchow & Lommatzsch, 2016*).

The MDSCs, a type of immature myeloid cells, are known for their immunosuppressive properties (*Hegde, Leader & Merad, 2021*). Predominantly, MDSCs are further classified into granulocyte-like MDSCs (G-MDSCs) and MNP-like MDSCs (M-MDSCs) based on their morphology and specific surface molecules (*Hegde, Leader & Merad, 2021*). The suppressive capacity of MDSCs in inhibiting the expansion of Tregs is

well-documented (*Veglia, Perego & Gabrilovich, 2018*; *Hegde, Leader & Merad, 2021*). Numerous studies have shown that MDSCs significantly impair adaptive antitumor immunity and are closely associated with unfavorable clinical outcomes in cancer (*Veglia, Perego & Gabrilovich, 2018*; *Veglia, Sanseviero & Gabrilovich, 2021*). However, the role of MDSCs was relatively inadequate in the case of non-neoplastic diseases, such as COPD. Additionally, the potential impact of MDSC subsets on Treg in COPD remains not yet fully understood.

Therefore, our study aimed to investigate the clinical significance of PD-1/PD-L1 axis expression, Tregs, and myeloid-derived cell subsets in the peripheral blood of COPD patients in order to explore the immuneregulatory role of MDSCs in COPD.

## MATERIALS AND METHODS

### Study design and patients

A total of 54 stable COPD patients spirometric stages I–IV (confirmed according to the Global Initiative for Obstructive Lung Disease (GOLD)) and 24 age-matched donors with normal lung function considered controls were enrolled. COPD was defined as follows: a post-bronchodilator FEV1/FVC ratio < 0.7 and FEV1 of less than 80% of the predicted value. COPD subjects with exacerbations within the 2-month prior to the study were excluded. Other exclusion criteria included the following: cancer, asthma, heart disease, autoimmune diseases, infectious diseases and administration with immunomodulatory drugs. Ethics committee approval (No. 18190-0-01) was obtained for the trial protocol at Tsinghua Changgeng Hospital. The remaining samples obtained from the patients after testing were utilized, and as a result, the ethics committee granted a waiver of informed consent for their use (No. 18190-0-01).

### Cell collection

Fresh peripheral blood samples were isolated from each subject and subjected to Ficoll-Paque Plus (GE Healthcare, Pittsburgh, PA, USA) layering and centrifugation (200 × 6 g for 5 min). Through Ficoll-Paque gradient centrifugation, human peripheral blood mononuclear cells were subsequently isolated.

### Flow cytometry analysis

Human PBMC were freshly obtained and stained with fluorochrome (APC, FITC, PE, PerCP-Cyanine5.5)-conjugated antibodies. For surface staining, the following antibodies were used: CD25 (cat. 12-0257-42), PD-1 (cat. 61-2799-42), PD-L2 (cat. 25-9952-42), CTLA-4 (cat. 85-46-1529-42), CD4 (cat. 25-0049-42), CD127 (cat. 17-1278-42), CD14 (cat. 61-0149-42), HLA-DR (cat. 25-9952), CD14 (cat. 61-0149-42), PD-L1 (cat. 85-12-5888-42), CD15 (cat. 11-0159-42), CD33 (cat. 56-0338-42), CD3 (cat. MHCD0327), CD11b (cat. 46-0118-42) for 30 min at 4 °C according to surface marker staining of each subpopulation and the corresponding isotype-matched controls were also used. eBioscience provided all of the antibodies we used. The gated strategies to identify Treg (CD3$^+$CD4$^+$CD25$^+$CD127$^{-/low}$), G-MDSCs (CD15$^+$CD33$^+$ CD11b$^+$CD14$^-$HLA-DR$^{-/low}$) and M-MDSCs (CD14$^+$ CD15$^-$ CD11b$^+$CD33$^+$ HLA-DR$^{-/low}$) were shown respectively in

**Table 1 Participants' clinical characteristics and demographics.**

| Variables | Healthy controls | COPD |
|---|---|---|
| Subjects (No.) | 24 | 54 |
| Age (year) | 73.6 ± 7.3 | 74.5 ± 8.2 |
| Gender (male/female) | 17/7 | 36/18 |
| Current/Ex-smokers | 5/11 | 14/24 |
| Smoking history (pack-year) | 21.80 (0–100) | 26.18 (0–100) |
| FEV 1 (% of predicted) | – | 51.47 ± 18.40 |
| FEV 1/FVC (%) | – | 51.15 ± 13.98 |
| Oral corticosteroid use | 0 | 0 |

Notes:
The data are presented as mean ± standard error of mean (SEM) or mean (range).
Abbreviations: COPD, patients with stable chronic obstructive pulmonary disease; FEV 1, forced expiratory volume in one second; FVC, forced vital capacity.

the below section. The data in this study were acquired using CytoFLEX instrument (Beckman Coulter, Pasadena, CA, USA) and analyzed using CytExpert for DxFLEX (Beckman Coulter, Pasadena, CA, USA).

## Statistical analysis

Data are expressed as the mean ± SEM. Comparisons between multiple groups were performed by variance (ANOVA) analysis, while statistical analysis was performed with Student's t-test between two groups. An evaluation of the correlation was made using Spearman' s rank correlation coefficient. Data were analyzed using GraphPad PRISM software (Version 8.0 for Windows; San Diego, CA, USA). Statistical significance was determined at a threshold of $p < 0.05$ (* for $p < 0.05$,** for $p < 0.01$, *** for $p < 0.001$, **** for $p < 0.0001$).

## RESULT

### Participants' clinical characteristics and demographics

A total of 54 COPD patients (25 in stages I–II and 29 in stages III–IV) and 24 healthy individuals participated in the study from September 9, 2021, to May 1, 2022. The clinical features of all the participants are provided in Table 1. In comparison with healthy subjects, COPD patients had no significant differences in age or smoking history. The sex ratio was unequal mostly because the incidence rate of COPD is higher among males than females in China.

### The frequency of MDSCs subsets in peripheral blood was significantly increased in COPD patients

To assess the role of MDSCs, we measured the proportion of two different subsets of MDSCs (M-MDSC and G-MDSC). The results depicted in Fig. 1 demonstrated a significantly higher proportion of M-MDSC in the peripheral blood of COPD patients compared to controls controls (9.02 ± 0.63 *versus* 16.33 ± 1.12, $p < 0.0001$) (Figs.1C and 1D). Additionally a greater diversity of G-MDSC percentages was observed in the

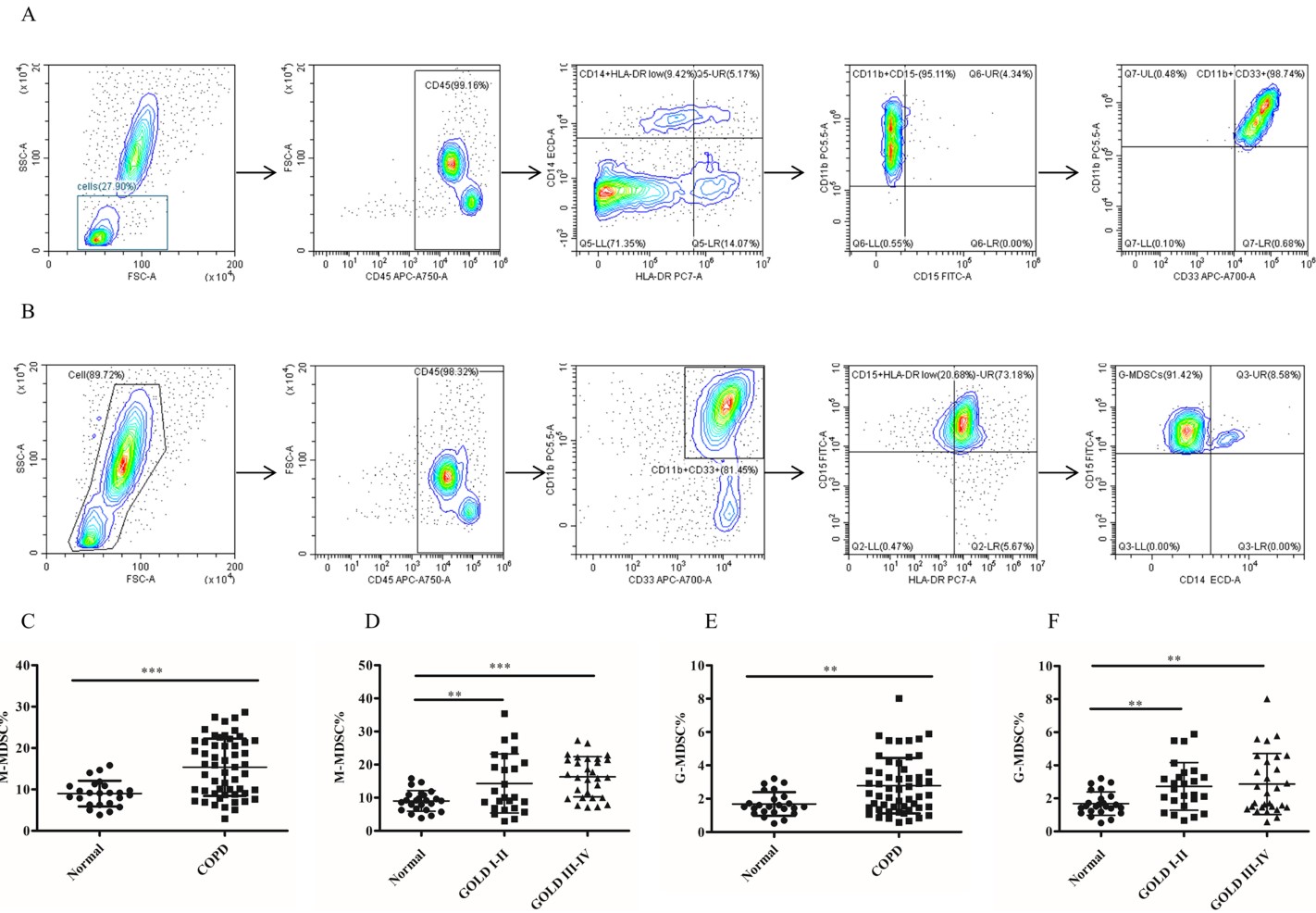

**Figure 1** **The frequency of MDSCs subsets in peripheral blood was significantly increased in COPD patients.** (A) The gated strategies to identify M-MDSCs; (B) Gated strategies for identifying G-MDSCs; (C and D) a higher diversity of M-MDSC percentages was observed in the peripheral blood of COPD patients compared with controls ($n$ = 54 and 24, respectively); (E and F) the percentages of G-MDSCs in peripheral blood of COPD patients were higher than those in control ($n$ = 54 and 24, respectively). Data were expressed as mean ± SEM, **$p$ < 0.01; ***$p$ < 0.001.

peripheral blood of COPD patients compared with controls (1.68 ± 0.14 *vs.* 2.67 ± 0.21, $p$ < 0.01) (Figs.1E and 1F).

## The frequency of Tregs and the CTLA-4 expression on Tregs in COPD patients peripheral blood

To investigate whether MDSC recruitment promotes Treg expansion in patients with COPD, we proceeded to analyze the frequency of circulating Tregs in their blood. As shown in Fig. 2B, there was a significantly increased proportion of Tregs (7.26 ± 0.29 *versus* 9.07 ± 0.38, $p$ < 0.01) in COPD patients peripheral blood in comparison with controls (Fig. 2B). Furthermore, the frequencies of Tregs were found to be elevated in COPD patients in GOLD stage I/II or stage III/IV on Tregs compared with matched healthy subjects (Fig. 2C).

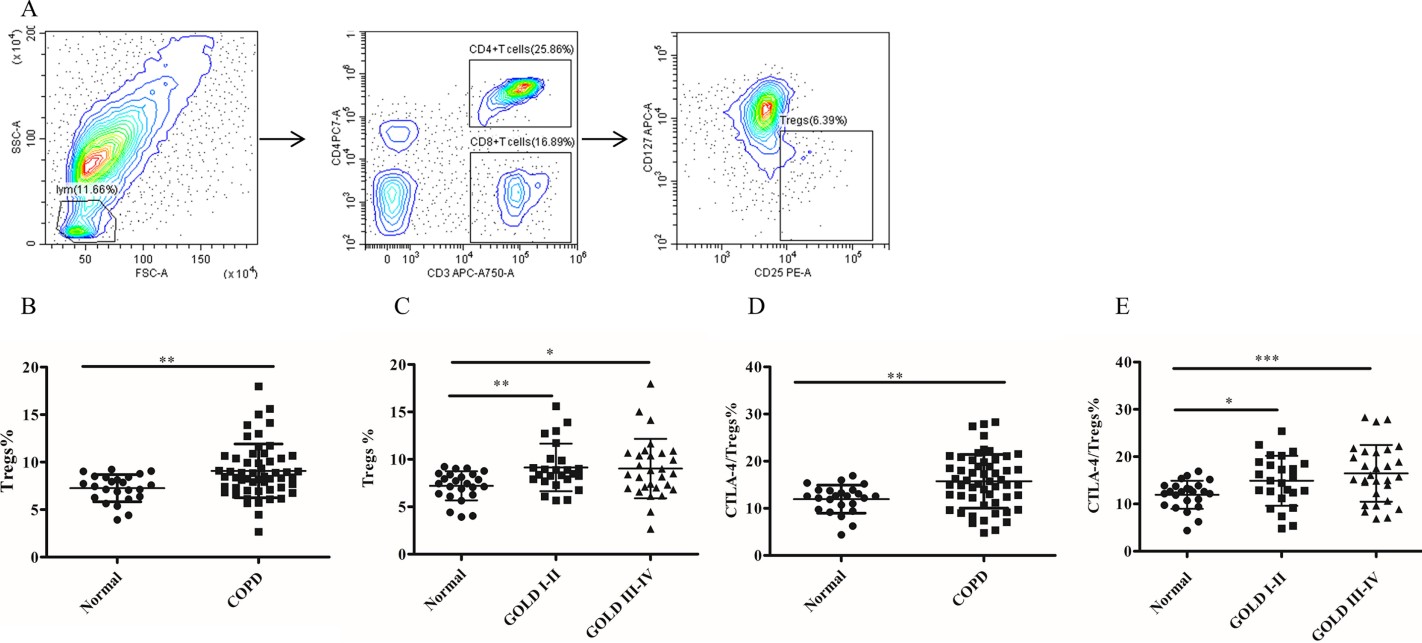

**Figure 2 The frequency of Tregs and the CTLA-4 expression on Tregs in COPD patients peripheral blood.** (A) The gated strategies for Tregs; (B and C) The increased percentage of Tregs in COPD patients peripheral blood compared with controls ($n$ = 54 and 24, respectively); (D and E) the increased expression of CTLA-4 on Tregs in COPD patients compared with controls ($n$ = 54 and 24, respectively). Data were expressed as mean ± SEM, *$p$ < 0.05; **$p$ < 0.01; ***$p$ < 0.001.                              

It has been established that CTLA-4 is essential for Treg-mediated immunosuppression and Tregs expressing CTLA-4 demonstrate enhanced immunosuppressive properties. The expression of CTLA-4 (11.96 ± 0.61 *versus* 15.74 ± 0.77, $p$ < 0.01) was significantly elevated on Tregs of COPD patients (Fig. 2D), indicating a heightened suppressive function of Tregs in COPD. Furthermore, the expression of CTLA-4 were upregulated in COPD patients in GOLD stage I/II or stage III/IV on Tregs compared with matched healthy individuals (Fig. 2E). However, no significant difference in CTLA-4 expression on Tregs was observed between GOLD stages I/II and stages III/IV COPD patients (Fig. 2E).

## The abnormal expression pattern of PD-L1 but not PD-L2 on MDSCs subsets in COPD patients

Although the role of MDSCs-mediated T-cell suppression under neoplastic disease has been extensively studied, their function and mechanism have not been fully elucidated in COPD. MDSCs exerted their immunosuppressive activities by expression or secretion of mediators or molecules, including PD-L1/PD-L2, which can lead to T cell exhaustion. In this study, we investigated the expression of PD-L1 and PD-L2 on different subsets of MDSCs using anti-CD274 (PD-L1) and anti-CD273 (PD-L2) antibodies. We detected PD-L1/PD-L2 expression on MDSC subsets. There was increased upregulation of PD-L1 on M-MDSCs in COPD patients compared to the control group (68.07 ± 4.67 *versus* 89.27 ± 1.93, $p$ < 0.0001, Figs. 3B and 3F). In contrast, significantly decreased expression of PD-L1 by G-MDSCs was seen in COPD patients blood compared with control group (70.59 ± 4.48 *versus* 53.74 ± 2.68, $p$ < 0.001, Figs. 3D and 3H). There was no statistically significant

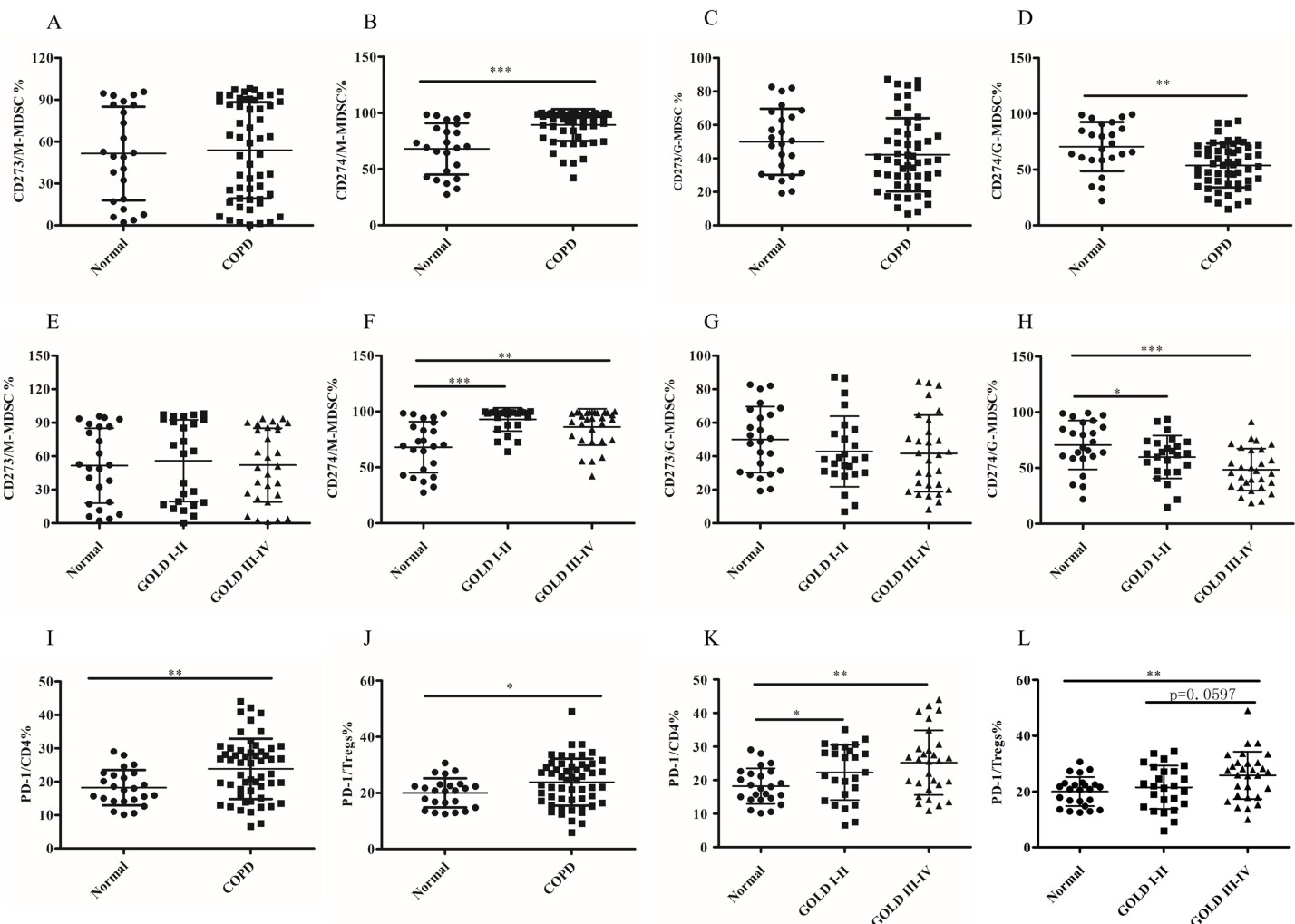

**Figure 3 The abnormal expression patternof PD-L1 on MDSCs subsets and the expression of PD-1 on T cell subsets in COPD patients peripheral blood.** (A and E) The expression level of CD273 (PD-L2) on M-MDSC in COPD patients peripheral blood; (B and F) the expression level of CD274 (PD-L1) on M-MDSC in COPD patients peripheral blood; (C and G) the expression level of CD273 (PD-L2) on G-MDSC in COPD patients peripheral blood; (D and H) the expression level of CD274 (PD-L1) on G-MDSC in COPD patients peripheral blood; (I and K) the expression of PD-1 on CD4[+]T cells in COPD patients peripheral blood; (J and L) the expression of PD-1 on Treg in COPD patients peripheral blood; Data were expressed as mean ± SEM, $^*p < 0.05$; $^{**}p < 0.01$; $^{***}p < 0.001$. $n = 54$ and 24 for COPD patients (25 in stages I–II and 29 in stages III–IV) and controls respectively.

difference observed in the expression level of PD-L2 between M-MDSC (Figs. 3A and 3E) or G-MDSCs (Figs. 3C and 3J) between the COPD and control groups.

## The expression of PD-1 on T cell subsets in COPD peripheral blood

The expression of PD-1, the receptor of PD-L1/L2, was assessed on CD4[+]T cells, CD8[+]T cells and Treg cells. Compared to the controls, the expression pattern of PD-1 was higher on CD4[+]T cells ($18.23 \pm 1.08$ versus $23.87 \pm 1.23$, $p < 0.001$; Fig. 3I) and Tregs ($20.02 \pm 1.06$ versus $23.84 \pm 1.14$, $p < 0.05$; Fig. 3J), but not on CD8[+]Tcells (data not shown). Next, subgroup analysis was performed based on the degree of airflow obstruction according to the Global Initiative for Chronic Obstructive Lung Disease (GOLD) guidelines. It is

noteworthy that COPD patients in GOLD stages I/II or stages III/IV exhibited a significant upregulation of PD-1 expression Ion CD4[+]Tcells, but not CD8[+]Tcells, compared to the control group (Fig. 3K). The expression of PD-1 on Treg showed a slight increase in patients with COPD in stages III/IV compared to those in stages I/II, although this difference was not statistically significant (21.53 ± 1.55 *versus* 25.83 ± 1.58, $p = 0.0597$; Fig. 3L).

## Relationship of MDSCs subsets, Tregs and PD-1/PD-L1 expression in COPD peripheral blood

The percentages of M-MDSCs cells were found to be positively correlated with the expression of PD-1 on CD4[+]T cells ($r = 0.42$, $p = 0.001$; Fig. 4A) and CD8[+]T cells ($r = 0.36$, $p = 0.008$) (Fig. 4B). Furthermore, upon further analysis, it was discovered that there was a positive correlation between the percentages of M-MDCS and the expression of PD-1 ($r = 0.51$, $p < 0.0001$; Fig. 4C) and CTLA-4 ($r = 0.42$, $p = 0.0014$; Fig. 4D) on Tregs. We also found a positive relationship between the percentages of M-MDSCs and Treg cells in COPD patients ($r = 0.315$, $p = 0.02$; Fig. 4E).

Notably, the percentages of G-MDSCs cells were positively related to the percentages of M-MDSCs ($r = 0.39$, $p = 0.0037$) (Fig. 4F). Our findings further demonstrated that the increased percentages of G-MDSCs was associated with the increased expression of PD-1 on CD4[+]T cells ($r = 0.299$, $p = 0.028$; Fig. 4G) and PD-1 on Treg ($r = 0.297$, $p = 0.028$; Fig. 4H). However, the proportions of G-MDSC or M-MDSC failed to significantly correlate with lung function in COPD patients.

## Relationship between pack year of cigarette smoking and the frequency of MDSCs subsets, Tregs and PD-1/PD-L1expression

The risk of COPD increases with pack-years of smoking. In our study, we observed a negative correlation between the number of pack-years of smoking and the expression of PD-1 on CD4[+]T cells ($r = -0.33$, $p = 0.04$; Fig. 4I), Tregs ($r = -0.37$, $p = 0.02$; Fig. 4J), and percentages of G-MDSCs cells ($r = -0.35$, $p = 0.03$; Fig. 4K) in COPD patients, but not the percentages of M-MDSCs, PD-L1 expression on MDSCs subsets and CTLA-4 expression on Treg.

## DISCUSSION

It is widely recognized that MDSCs have the ability to inhibit the expansion and activation of Tregs in various pathological conditions (*Hegde, Leader & Merad, 2021*). A sufficient understanding about the role of MDSC subsets and potential relationships between MDSC subsets and Tregs in COPD has not yet been achieved. In this study, we observed significantly elevated percentage of circulating M-MDSCs displayed a high degree of positive correlation with suppressive Tregs in COPD, accompanied by a significantly increased expression of PD-1/PD-L1 axis on specific immune cells.

MDSCs have been extensively studied in patients with cancer, but there is a lack of research on their role in non-malignant conditions, particularly in COPD. Some researchers reported that circulating MDSCs were up-regulated in COPD (*Scrimini et al.,*

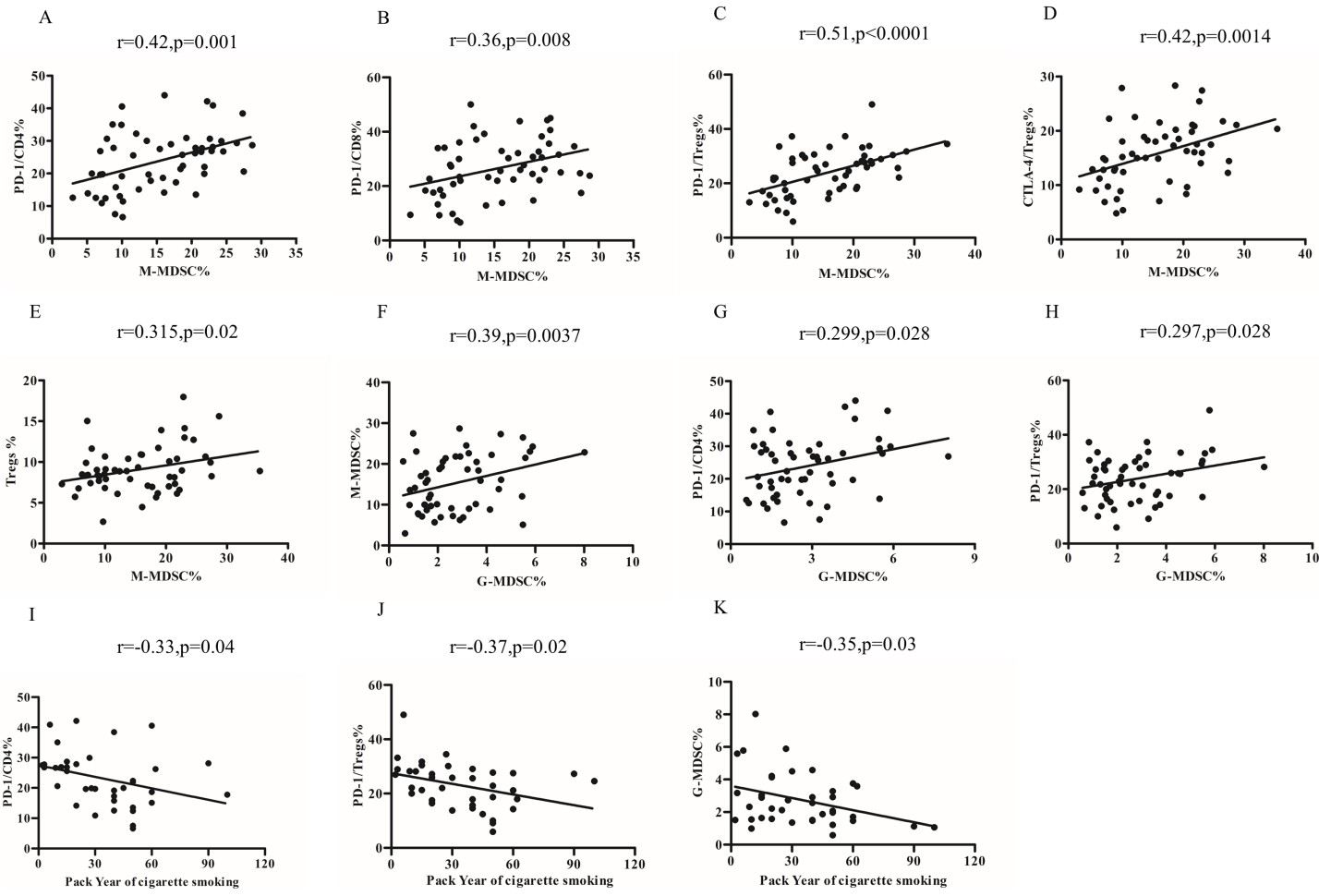

**Figure 4 Relationship of MDSCs subsets, Tregs and PD-1/PD-L1 expression in COPD peripheral blood.** (A and B) The percentages of M-MDSCs cells were positively correlated with the expression of PD-1 on CD4$^+$T cells (A, r = 0.32, p = 0.001, n = 54) and CD8$^+$T cells (B, r = 0.36, p = 0.008, n = 54); (C and D) the percentages of M-MDCS were positively correlated with the expression of PD-1 on Treg (C, r = 0.36, p = 0.008, n = 54) and CTLA-4 on Tregs (D, r = 0.42, p = 0.0014, n = 54); (E) the percentages of M-MDCS were positively correlated with the percentages of M-MDSCs and Treg cells in COPD patients (r = 0.315, p = 0.02, n = 54); (F) the percentages of G-MDSCs cells were positively related to the percentages of M-MDSCs in COPD patients (r = 0.29, p = 0.028, n = 54); (G and H) increased percentages of G-MDSCs was associated with the increased expression of PD-1 on CD4$^+$T cells (G, r = 0.299, p = 0.028, n = 54) and Treg (H, r = 0.297, p = 0.028, n = 54) and in COPD patients; (I-K) Relationship between smoking pack-years and PD-1 expression on CD4$^+$T cells (I), PD-1 expression on Tregs (J, r = −0.33, p = 0.04, n = 54), and percentages of G-MDSCs (K, r = −0.35, p = 0.03, n = 54) in COPD peripheral blood.           

2013; *Kalathil et al., 2014*; *Deshane et al., 2015*; *Scrimini et al., 2015*; *Wu et al., 2015*). However, another study reported that COPD was not associated with perturbations in the percentages or the function of MDSCs in peripheral blood (*Tan et al., 2014*). Furthermore, there is limited investigation into the subsets of MDSCs in COPD patients. In our study PBMCs from COPD contained both an elevated frequency of G-MDSCs and M-MDSCs compared to controls group, which is in line with previous on MDSCs (*Scrimini et al., 2013*; *Kalathil et al., 2014*; *Deshane et al., 2015*; *Scrimini et al., 2015*; *Wu et al., 2015*).

Treg, a subpopulation of CD4$^+$T cells with immunosuppressive function, plays a crucial role in the suppression of inflammatory pathology. Previous studies have yielded conflicting results regarding the specific role of Treg in COPD. *Chiappori et al. (2016)*

demonstrated lower numbers of circulating Treg cells in COPD patients. According to another study, smokers with COPD also have significantly fewer Treg cells in their lungs (*Lee et al., 2007*). Conversely, accumulating evidence shows that the frequency of Tregs is increased both in mice exposed to smoke (*Gong et al., 2017*) and COPD patients (*Plumb et al., 2009*; *Kalathil et al., 2014*). Consistent with these findings, our study revealed that the proportion of circulating Treg cells in COPD patients was increased compared with the control groups (*Kalathil et al., 2014*) and showed an positive correlation with the percentage of M-MDSCs, consistent with studies that suggest MDSCs may be crucial to Treg expansion (*Kalathil et al., 2014*).

Previous research has established the critical role of CTLA-4 in the development and suppressive function of Tregs (*Zheng et al., 2006*; *Wing et al., 2008*) and CTLA-4+ Tregs have increased suppressive capacity (*Read, Malmström & Powrie, 2000*; *Núñez et al., 2020*). Our result indicate that there is an increased expression of CTLA-4 on Tregs in COPD patients, and a positive correlation between the percentage of M-MDSCs and the expression of CTLA-4 on Tregs. This suggests that MDSCs may also play a role in modulating the suppressive function of Tregs.

The PD-1/PD-L1(L2) signaling is known to regulate immune tolerance in many immune-mediated diseases, including cancer, autoimmune diseases, and, as emphasized recently, in chronic inflammation and regulates T cell activation negatively and regulates the generation and function of Treg (*Adamczyk & Krasowska, 2021*; *Filippone et al., 2022*). PD-1 is expressed on Tregs and partially proved to represse Treg suppressive function (*Kumagai et al., 2020*; *Lowther et al., 2016*) and high PD-1 expression on Treg cells indicates enhanced Treg function (*Asano et al., 2017*). Previous research has demonstrated that treatment with anti-PD-1 can mitigate lung damage and neutrophilic inflammation (*Ritzmann et al., 2021*) and acute exacerbations of COPD (*Tan et al., 2018*), suggesting the involvement of the PD-1 axis in the pathogenesis of COPD. *McKendry et al. (2016)* discovered that the expression of PD-1 on lung CD8+T cells was elevated in individuals with COPD compared to controls. Similarly, our study revealed a higher level of PD-1 expression on CD4+ T cells and Tregs, but not on CD8+ T cells, in GOLD stage I/II and stage III/IV COPD patients compared to controls. These findings align with previous research.

Based on previous research, it has been observed that, MDSCs in tumor-bearing mice may exhibit increased expression of PD-L1 under hypoxia (*Noman et al., 2014*). Furthermore, chronic inflammation induced by LPS has been found to up-regulate the PD-1/PD-L1 axis and lead to the accumulation of MDSCs (*Liu et al., 2021*). Additionally, a study conducted by *Rui et al. (2022)* revealed that exposure to cigarette smoke results in the upregulation of PD-L1 in rats with COPD. There is limited information about PD-L1/PD-L2 expression in MDSC subsets in COPD as far as we know. In line with these results, we found the up-regulationof PD-L1 but not PD-L2 on M-MDSCs subsets in COPD patient peripheral blood (*Kalathil et al., 2014*; *Liu et al., 2021*). Interestingly, a notable decrease in the expression of PD-L1 by G-MDSCs was observed in patients with COPD. This finding leads us to suspect that the functionality of G-MDSCs may be more severely impaired in individuals with COPD.

Smoke is the most important risk factor for COPD. However, the relationship between smoking and MDSC subset frequency in COPD remains poorly understood. Worth mentioning, pack-years of smoking were negatively correlated with the percentages of G-MDSCs cells in our study and imply an inducible decrease in the number of G-MDSCs by smoking. *Matsuda et al. (2018)* revealed that smoking suppresses PD-1 expression and limited serum sPD-L1 in rheumatoid arthritis (RA) patients (*Luo et al., 2018*). It has, however, been suggested an inducible expression of PD-L1 by smoking (*Calles et al., 2015*; *Psomas et al., 2019*). Our findings indicate a negative correlation between pack-years of cigarette smoking and the expression of PD-1 on Tregs and CD4$^+$ T cells. The conflicting conclusions could potentially be attributed to the varying roles of the PD-1/PD-L1 axis in different conditions.

As in other studies, the present work also has some limitations. Firstly, the present study is descriptive in nature and confirmation of this explanation will requires further experiments *in vitro*. Secondly, despite our interesting findings, the relatively small sample size and single center design may restrict the generalizability of our results. Additionally, certain risk factors such as physical exercise and dietary intake were not taken into account, potentially influencing the relationship between pack year of cigarette smoking and our findings. Fourthly, the fixation and membrane rupture of cells are required for the detection of Foxp3, which may lead to the destruction and death of lymphocytes. Therefore, Treg cells were characterized by CD3$^+$CD4$^+$CD25$^+$CD127$^{-/\text{low}}$ in our study. If FMO controls is used as negative controls, in place of isotype controls, the results of our study would be more accurate. Therefore, caution should be exercised when interpreting the association between these variables.

## CONCLUSIONS

In conclusion, we observed the accumulation of circulating MDSCs subsets, immunosuppressive Tregs and upregulation expression of the PD-1/PD-L1, not PD-L2, on these two cell populations in COPD patients and we speculated that PD-1/PD-L1 axis may be involved in MDSCs, especially M-MDSCs, induced Tregs expansion and activation at least partially in COPD patients. Additionally, smoking status was inversely related to the percentage of G-MDSCs and decreased expression of PD-L1 by G-MDSCs, implying the negative effects of smoking on G-MDSCs cells production and the impaired immunosuppressive function of G-MDSCs. However, our result need more experiments *in vitro* and *in vivo* to prove. A comprehensive and in-depth understanding of MDSC subsets in COPD may help to provide useful prognostic indicators and effective therapeutic targets for the disease.

### Funding

This work was supported by grants from the National Natural Science Foundation of China (no. 81900021, no. 82000048) and the Beijing Clinical Key Specialty (grant no.

XKB2022B1002). The funders had no role in study design, data collection and analysis, decision to publish, or preparation of the manuscript.

## Grant Disclosures

The following grant information was disclosed by the authors:

National Natural Science Foundation of China: 81900021, 82000048.

Beijing Clinical Key Specialty: XKB2022B1002.

## Competing Interests

The authors declare that they have no competing interests.

## Author Contributions

- Mingqiang Zhang conceived and designed the experiments, performed the experiments, analyzed the data, prepared figures and/or tables, authored or reviewed drafts of the article, and approved the final draft.
- Yinghua Wan performed the experiments, analyzed the data, prepared figures and/or tables, authored or reviewed drafts of the article, and approved the final draft.
- Jie Han performed the experiments, prepared figures and/or tables, authored or reviewed drafts of the article, and approved the final draft.
- Jun Li performed the experiments, prepared figures and/or tables, authored or reviewed drafts of the article, and approved the final draft.
- Haihong Gong conceived and designed the experiments, performed the experiments, analyzed the data, prepared figures and/or tables, authored or reviewed drafts of the article, and approved the final draft.
- Xiangdong Mu performed the experiments, analyzed the data, authored or reviewed drafts of the article, and approved the final draft.

## Human Ethics

The following information was supplied relating to ethical approvals (*i.e.*, approving body and any reference numbers):

Ethics committee approval was obtained for the trial protocol at Tsinghua Changgeng Hospital (No. 18190-0-01).

## Ethics

The following information was supplied relating to ethical approvals (*i.e.*, approving body and any reference numbers):

Ethics committee approval was obtained for the trial protocol at Tsinghua Changgeng Hospital.

## Data Availability

The raw measurements are available in the Supplemental File.

## Supplemental Information

Supplemental information for this article can be found online at http://dx.doi.org/10.7717/peerj.16988#supplemental-information.

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
