# Peer review of "The clinical association of programmed death-1/PD-L1 axis, myeloid derived suppressor cells subsets and regulatory T cells in peripheral blood of stable COPD patients"

_PeerJ, doi:10.7717/peerj.16988_

## Round 0.1 · original submission · Major Revisions

Reviewers have some issues with your manuscript.

Please review in detail and respond in consequence. Note that some reviewers have recommended improving the English language.

**Language Note:** The Academic Editor has identified that the English language must be improved. PeerJ can provide language editing services - please contact us at [email protected] for pricing (be sure to provide your manuscript number and title). Alternatively, you should make your own arrangements to improve the language quality and provide details in your response letter. – PeerJ Staff

·

Basic reporting

1. The presentation of the text, figures, and figure legends must be improved in quality and presentation. For example, the figures must homologate the lines of significance between groups. Figures 2, 3, and 4 are shown with superimposed panels and texts, which are not legible and displayed entirely out of order (unlike Figure 1). This should not happen, even in the manuscript revision stage.

2. The figure legends must also be approved in the content of the information. In particular, Figure Legend 4 presents a small description of the observed correlations, but it needs to be included for panel E. Additionally, it is suggested that the figure legend describes the statistical analysis carried out, the level of significance for all cases, the ‘n’ of each represented group, and also Rho for correlations.

3. The English must be improved. Likewise, space tabulations need to be included throughout the text.

4. Although an English version file from the Ethic Committee Approval is included, there is no marking that this is an official translation. In any case, it is a requirement that the Journal's editorial team must decide if it is necessary.

Experimental design

The work was carried out with a clear and straightforward experimental analysis that provides an interesting description of the identified cell populations. However, some aspects must be clarified.

1. Did the authors pre-tested for normal distribution of the data? Please describe to clarify the selection of the statistical analysis you performed.

2. Did the authors analyze the prevalence of comorbidities in COPD patients? As epidemiological studies have shown that COPD is frequently associated with comorbidities, it is essential to mention the prevalence in the studied population and include it in Table 1.

3. Additionally, the study considered the analysis of smoking association with the findings; however, other risk factors, such as diet and physical activity, could be relevant. Please comment if these factors were considered. If so, they should be included in Table 1. If not, please justify.

Validity of the findings

In the research from Mingqiang Zhang et al., the authors focus on the clinical significance of PD-1/PD-L1 axis expression in the accumulation of circulating MDSCs subsets, especially M-MDSCs, induced Tregs expansion and activation in COPD patients.

1. Although the study focuses on Treg cells, did the authors identify any other CD4+ T cell population that may be relevant? This could be interesting because, as the authors present in Figure 3I, PD1 is increased in the overall CD4+ population. So, this molecule is not exclusively overexpressed in Tregs. Please describe to clarify the discrimination of other CD4+ populations in the study.

2. As has been broadly described in several reports, Tregs are cells with dual roles in the modulation of immune responses, so this population does not present only a negative regulatory activity, as has been affirmed in line 219 of the manuscript.

3. The authors conclude that myeloid-derived suppressor cell recruitment, Treg accumulation, and up-regulation of CTLA-4 on Treg in COPD, accompanied by an increased level of PD-1/PD-L1, suggest MDSCs may induce the expansion of highly suppressive Treg cells through the PD-1/PD-L1 pathway.

The previous is in agreement with the observations done. However, do the authors have the possibility to evaluate any other immune inhibitory factor that may lead to a more specific mechanism in MDSCs? For example, when activated in a pathogenic situation, MDSC cells overexpress immune inhibitory factors such as nitric oxide synthase, arginase 1, and peroxynitrite. Please justify.

4. At the end of the introduction, the authors promise to associate the experimental results with the clinical significance. However, it is a promise that research does not fulfill. Beyond analysis in patient groups, a correlation of the speculated mechanism with clinical data is needed. This could be enriched with a figure that integrates the findings, in which cell populations, regulatory molecules, possible cellular mechanisms triggered, and their impact at a clinical level should be interrelated.

Additional comments

The work requires improving the quality of its presentation in the text, figures, table, and figure legends. Although the editorial team must supervise it, the authors should have previously supervised it.

An integrative idea is also necessary to describe possible molecular mechanisms that may be involved in both cell populations according to the molecules identified.

·

Basic reporting

Zhang M et al. studied the immunomodulatory role of MDSCs on the Treg in COPD patients. The authors concluded that COPD patients have an accumulation of MDSC, Treg CTLA4+, and high PD-1/PD-L1 levels, suggesting that MDSCs induce the expansion of Treg through the PD-1 pathway.
This study is relevant to understanding the immunological changes during COPD; the question in this study is well planned. However, the methodological design needs to be revised. Thus, I have major and minor comments.

Experimental design

Mayor comments

1. Add a table as supplementary where you indicate each antibody is conjugate to which fluorochrome, company, and clone. Moreover, indicate which viability sonde you used and justify why you used isotype control and no FMO. This last point is important, mainly to identify high and low expression.

2. According to methods, you used FACs Calibur, and this machine has the maximum capacity to identify 6 colors. Figure 1 shows that you needed at least 6 colors to identify MDSC subsets. How did you use an extra color to evaluate viability or PDL molecules?

3. To identify Treg (Figure 2), should be used the phenotype: CD2+ CD3+ CD4+ CD25+ FoxP3+. Why did you not use FoxP3? CD127 could be excluded, but FoxP3 is the fundamental marker of Treg.

4. I suggest measuring TNF/TNFRs to evaluate possible correlations because there is a lot of evidence that TNF blocks the differentiation of MDSC and Treg (although this population is more dependent on TNFR2). You can obtain more references about this topic at doi.org/10.3389/fimmu.2017.00999. Considering the TNF effect on these cell subsets is necessary because it is one of the most classical and common cytokines that is increased under inflammatory diseases

5. Both the title and conclusion overestimate the obtained results. This study is only descriptive; you have not demonstrated (with blockers antibody or know down, know out cells) that the block of the PD pathway controls the expansion and activation of MDSC and Treg. Thus, the relation you found (figure 4) between cell subsets and PDL molecules has a good p-value, but the r-value is weak. Consequently, more is needed to conclude the real impact of PD/PDL on cells; you need to develop experiments (as suggested in this comment) to confirm the proposed conclusion.

Validity of the findings

Minor comments

1. Abs (line 111) has not indicated means (I assume it is an antibody).
2. I suggest including (maybe as supplementary) a complete strategy analysis of flow cytometry to clarify how you develop the follow of the gate by the gate to each subpopulation and the molecules you evaluated into the subpopulation. It could be helpful to avoid confusion about the machine's efficiency in developing the analysis where you need more than 6 colors.
3. The discussion section can be improved.

·

Basic reporting

In this work, Mingqiang Zhan et al. evaluated the expression of immune checkpoints, including CTLA-4, PD-1, and their ligands PD-L1/2 in M-MDSCs and G-MDSCs obtained from COPD patients and compared them with healthy donors. They found increased levels of MDSCs, Tregs CTLA-4+, T cells CD4+, PD-1+, and Tregs PD-1+ in COPD patients compared to age-matched healthy donors.

The study was conducted correctly, and the results are well-presented. However, they mentioned results about CD8+ T cells not included in Figure 3. I would like to ask if the omission of this graphic is due to the measurement of CD8+ T cells.

Experimental design

The experimental design of this research was performed without complications, and the results show solid data.

Validity of the findings

The results are well presented and accurately reflect the immune cell parameters of COPD patients and healthy donors.

Additional comments

The authors have made the following supposition:

'Myeloid-derived suppressor cell recruitment, Treg accumulation, and up-regulation of CTLA-4 on Treg in COPD, accompanied by an increased level of PD-1/PD-L1, suggest that MDSCs may induce the expansion of highly suppressive Treg cells through the PD-1/PD-L1 pathway.'

However, it is important to note that the role of the PD-1 axis in MDSCs is not entirely clear. The primary mechanism of PD-1 involves the participation of SHP phosphatases to inhibit T cell activation. Therefore, it is necessary to provide a more comprehensive explanation or suggestion regarding the relationship between PD-1 expression and the development of Tregs.

To enhance the discussion section and the conclusion, it would be beneficial to consider and discuss evidence regarding how PD-1 represses Treg suppressive function. This approach will allow the authors to present a more in-depth discussion.

Ref:
1-Lowther DE, Goods BA, Lucca LE, Lerner BA, Raddassi K, van Dijk D, Hernandez AL, Duan X, Gunel M, Coric V, Krishnaswamy S, Love JC, Hafler DA. PD-1 marks dysfunctional regulatory T cells in malignant gliomas. JCI Insight. 2016 Apr 21;1(5):e85935. doi: 10.1172/jci.insight.85935. PMID: 27182555; PMCID: PMC4864991.

2-Kumagai S, Togashi Y, Kamada T, Sugiyama E, Nishinakamura H, Takeuchi Y, Vitaly K, Itahashi K, Maeda Y, Matsui S, Shibahara T, Yamashita Y, Irie T, Tsuge A, Fukuoka S, Kawazoe A, Udagawa H, Kirita K, Aokage K, Ishii G, Kuwata T, Nakama K, Kawazu M, Ueno T, Yamazaki N, Goto K, Tsuboi M, Mano H, Doi T, Shitara K, Nishikawa H. The PD-1 expression balance between effector and regulatory T cells predicts the clinical efficacy of PD-1 blockade therapies. Nat Immunol. 2020 Nov;21(11):1346-1358. doi: 10.1038/s41590-020-0769-3. Epub 2020 Aug 31. PMID: 32868929.

3-Strazza M, Adam K, Lerrer S, Straube J, Sandigursky S, Ueberheide B, Mor A. SHP2 Targets ITK Downstream of PD-1 to Inhibit T Cell Function. Inflammation. 2021 Aug;44(4):1529-1539. doi: 10.1007/s10753-021-01437-8. Epub 2021 Feb 24. PMID: 33624224; PMCID: PMC9199348.


Other minor corrections are listed below:

Lane 33: "There is a mistake in the word 'expressionon'."

Lane 93: "1."

Lanes 159-161: "MDSCs exerted their immunosuppressive activities by the expression or secretion of mediators or molecules, including PD-L1/PD-L2, which could lead to T cell exhaustion." Reference necessary.

Lane 132: "2."

Lanes 161 and 162: "There are different ways to abbreviate both PD-1 ligands. For example, in line 161: 'PD-L1/PD-L2,' meanwhile in line 162: 'PD-L1(L2).' It would be clearer to choose only one form."

Lane 251: "up-regu1ation"

Result section: "There are many unnecessary repeated CD274 (PD-L1) and CD273 (PD-L2). Please leave this only the first time that it appears in the text, and the rest can be written as PD-L1 or PD-L2 since they are more common names for these receptors."

Lane 222: "In the whole manuscript, there are many parentheses that lack space, for example: 'patients(Chiappori et al., 2016). According.' Please review and correct this.

---

## Round 0.2 · Major Revisions

The authors should reply to reviewer 2's concerns since she comments about the wrong use of FMO.

Please consider incorporating the appropriate sentences in this matter into the manuscript either directly or in the limitations sections.

·

Basic reporting

Accept

Experimental design

Accept

Validity of the findings

Accept

Additional comments

A figure or scheme that integrates the findings of the work is necessary. The figure may represent which cell populations, regulatory molecules, and cellular mechanisms triggered, and their impact at a clinical level should be interrelated. This suggestion was performed in the first revision; however, the authors did not consider it. In any case, the editor can decide on the appropriateness of the suggestion.

·

Basic reporting

The authors did not reply to my main questions; from my viewpoint, this study needs to improve its experimental design.

Experimental design

The authors replied that isotope controls replaced FMO because other studies used that; however, the current flow cytometry rules indicate that FMO is better when you divide into a high or low expression because sometimes the fluorochromes have to overlap. The use of isotope controls is not useful in limiting the fluorescence overlapping.
Moreover, the authors sent me references to replace FoxP3 with CD127 as a marker of Treg; when I read all the references, the use of FoxP3 is maintained as the main marker in all those papers (and the guidelines to flow cytometry is irrelevant to determinate a phenotype, they are used to the adequate use of the flow cytometry, in fact, in the more recent version the authors can read about the use of FMO). From my viewpoint, authors not necessary are reported Treg, if they want use CD127 so the phenotype should me extended.

Validity of the findings

Authors need to validate their results.

·

Basic reporting

Mingqiang Zhan et al. evaluated the expression of immune checkpoint receptors, including CTLA-4, PD-1, and their ligands PD-L1/2, in M-MDSCs and G-MDSCs obtained from COPD patients, comparing them with healthy donors. The significant findings include increased levels of MDSCs, Tregs CTLA-4+, T cells CD4+, PD-1+, and Tregs PD-1+ in COPD patients compared to age-matched healthy donors.

The study was conducted correctly, and the manuscript has substantially improved considering the comments from reviewers 1 and 2.

Experimental design

The experimental design of this work was executed correctly to obtain valid results. Despite not including the nuclear factor FoxP3 in the Flow cytometry panel, it is well accepted that cells CD4(+)CD25(+)CD127(low/-) correspond to the phenotype of regulatory T cells (1).

1.- Yu N, Li X, Song W, Li D, Yu D, Zeng X, Li M, Leng X, Li X. CD4(+)CD25 (+)CD127 (low/-) T cells: a more specific Treg population in human peripheral blood. Inflammation. 2012 Dec;35(6):1773-80. doi: 10.1007/s10753-012-9496-8. PMID: 22752562.

Validity of the findings

The presentation of results in this study has significantly improved, featuring a well-structured narrative. This revised version enhances clarity in communicating the findings, highlighting the author's commitment to transparency and precision. The discussion section, following the results, is more measured and coherent.

The improved presentation of results, characterized by clarity and a more critical approach, enhances the credibility of the study's findings.

Additional comments

The authors have effectively addressed my observations and concerns.

---

## Round 0.3 · accepted · Accept

The authors have addressed all the reviewers' comments. Also, two previous reviewers have recommended accepting this manuscript; I agree with the current version and, in my opinion, the manuscript is ready for publication.